# Somatic Mutations in miRNA Genes in Lung Cancer—Potential Functional Consequences of Non-Coding Sequence Variants

**DOI:** 10.3390/cancers11060793

**Published:** 2019-06-08

**Authors:** Paulina Galka-Marciniak, Martyna Olga Urbanek-Trzeciak, Paulina Maria Nawrocka, Agata Dutkiewicz, Maciej Giefing, Marzena Anna Lewandowska, Piotr Kozlowski

**Affiliations:** 1Institute of Bioorganic Chemistry, Polish Academy of Sciences, 61-704 Poznan, Poland; paulingalka@gmail.com (P.G.-M.); martyna.urbanek@gmail.com (M.O.U.-T.); pnawrocka@ibch.poznan.pl (P.M.N.); agg.dutkiewicz@gmail.com (A.D.); 2Institute of Human Genetics, Polish Academy of Sciences, 60-479 Poznan, Poland; maciej.giefing@igcz.poznan.pl; 3The F. Lukaszczyk Oncology Center, Department of Molecular Oncology and Genetics, 85-796 Bydgoszcz, Poland; lewandowskam@co.bydgoszcz.pl; 4The Ludwik Rydygier Collegium Medicum, Department of Thoracic Surgery and Tumours, Nicolaus Copernicus University, 85-796 Bydgoszcz, Poland

**Keywords:** miRNA, somatic mutations, lung cancer, TCGA, non-coding

## Abstract

A growing body of evidence indicates that miRNAs may either drive or suppress oncogenesis. However, little is known about somatic mutations in miRNA genes. To determine the frequency and potential consequences of miRNA gene mutations, we analyzed whole exome sequencing datasets of 569 lung adenocarcinoma (LUAD) and 597 lung squamous cell carcinoma (LUSC) samples generated in The Cancer Genome Atlas (TCGA) project. Altogether, we identified 1091 somatic sequence variants affecting 522 different miRNA genes and showed that half of all cancers had at least one such somatic variant/mutation. These sequence variants occurred in most crucial parts of miRNA precursors, including mature miRNA and seed sequences. Due to our findings, we hypothesize that seed mutations may affect miRNA:target interactions, drastically changing the pool of predicted targets. Mutations may also affect miRNA biogenesis by changing the structure of miRNA precursors, DROSHA and DICER cleavage sites, and regulatory sequence/structure motifs. We identified 10 significantly overmutated hotspot miRNA genes, including the miR-379 gene in LUAD enriched in mutations in the mature miRNA and regulatory sequences. The occurrence of mutations in the hotspot miRNA genes was also shown experimentally. We present a comprehensive analysis of somatic variants in miRNA genes and show that some of these genes are mutational hotspots, suggesting their potential role in cancer.

## 1. Introduction

Lung cancer is the most common cause of cancer-related morbidity and mortality worldwide [1] and is defined as a group of distinct diseases with high genetic and cellular heterogeneity [2]. Non-small cell lung carcinoma (NSCLC) is the most common lung cancer subtype and can be further divided into lung adenocarcinoma (LUAD), lung squamous cell carcinoma (LUSC), and large cell carcinoma (LCC). Genetic profiles of these cancers have been revealed by several whole genome and whole exome next-generation sequencing (NGS) projects that led to the identification of thousands of somatic mutations within individual cancer genomes [3,4,5,6]. Analysis of these mutations allowed the elucidation of several important protein-coding driver genes, including *KRAS*, *EGFR*, *BRAF*, *MET*, *RIT1*, *ALK*, and *NF1* in LUAD and *PIK3CA*, *FGFR1*, and *PTEN* in LUSC [7,8,9]. Therapeutic strategies specifically targeting some of these drivers have been developed and successfully trialed, and they are now the most prominent examples of successful personalized/targeted therapies [10]. However, key drivers are not yet recognized for substantial fractions of LUAD and LUSC cases [7,8,9,10,11].

Other functional genetic elements coded by non-protein-coding sections of the genome include short non-coding single-stranded RNA particles called microRNAs (miRNAs). It is estimated that miRNAs regulate the expression of most protein-coding genes [12,13]. At present, nearly 2000 human miRNAs have been described, but the biological functions of most miRNAs remain unknown [14]. miRNA-coding sequences are not randomly distributed over the genome and are overrepresented in certain positions associated with fragile sites involved in cancer [15]. miRNAs may be encoded in independent transcriptional units or protein-coding genes in either the sense or antisense orientation and are mostly expressed as long 5′-capped and 3′-polyadenylated primary transcripts (pri-miRNAs). Mature miRNAs are generated in cells in a multistage process of miRNA biogenesis [16,17,18]. In the nucleus, pri-miRNAs are processed by the RNase DROSHA and DGCR8 within the microprocessor complex to release hairpin miRNA precursors (~80 nt, pre-miRNAs). After pre-miRNA export to the cytoplasm, the RNase DICER removes the apical loop to release a 19-25-bp miRNA duplex containing 2-nt 3′ overhangs on both ends. The miRNA duplex is then incorporated into the miRNA-induced silencing complex (RISC), where it is unwound; one strand (passenger strand) is released, and the other strand (guide strand or mature miRNA) is selected to target complementary transcripts. Generally, miRNAs function as cytoplasmic regulators via base-pairing with complementary (or nearly complementary) sequences within mRNA (mostly in the 3′UTR). This posttranscriptional silencing of gene expression occurs through transcript deadenylation and/or degradation or translation inhibition. Canonical miRNA:target interactions occur via complementarity of the 7-nt seed region defined by nucleotides 2-8 of mature miRNA. Additional mechanisms that regulate the level and fidelity of miRNA maturation and function exist at each step of miRNA biogenesis [16,19]. For example, specific structural features and primary sequence motifs (basal UG, CNNC, and loop UGUG motifs) present in miRNA precursors were shown to facilitate miRNA processing [20,21].

The important role of miRNA in the regulation of physiological processes such as growth, development, differentiation, proliferation, and apoptosis [22,23] prompted extensive studies in cancer. For example, dozens of miRNA expression profiling studies in lung cancer have been performed, and many consistently overexpressed (e.g., miR-21, miR-210, miR-182, miR-31, miR-200b, and miR-205) and underexpressed miRNAs (e.g., miR-126, miR-30a, miR-30d, miR-486, miR-451a, and miR-143) have been identified (for summary, see previous meta-analyses [24,25]). It has been determined that the upregulation or downregulation of certain miRNAs may contribute to carcinogenesis, and therefore, such miRNAs may be classified as either oncogenes (oncomiRs) or tumor suppressors (suppressormiRs) [26,27]. Among the most intensively studied oncomiRs in lung cancer and other types of cancer are miR-21, miR-155, the miR-17-92 cluster and miR-205. Similarly, a group of suppressormiRs, such as those in the let-7 and miR-200 families and miR-143, has been identified. miRNAs have been shown to play an important role in many oncogenic processes, including proliferation, epithelial-mesenchymal transformation (EMT), migration, angiogenesis, inflammation, apoptosis, and response to cancer treatment. Thus, miRNAs have been implicated as diagnostic and prognostic biomarkers and cancer therapeutic targets (reviewed in [28,29,30,31]). Additionally, miRNA genes are often either amplified or deleted in cancer in a similar fashion as protein coding oncogenes and tumor suppressor genes, and somatic copy number variation may be an important mechanism underlying aberrant miRNA expression in cancer [15,32,33].

In contrast to the excitement about the role of miRNA in cancer, very little is known about somatic mutations in miRNA genes (defined here as sequences encoding pre-miRNAs and their directly adjacent flanks; note that actual miRNA genes are much larger, encoding entire transcription units) in cancer. Important exceptions are (i) a recently published study reporting a tool (ADmiRE) for the annotation and prioritization of different genetic variants, including somatic mutations in miRNAs [34], (ii) a database of somatic mutations affecting the interactions of miRNAs with competing endogenous RNAs (SomamiR), including somatic mutations in miRNAs [35], and (iii) recent reports of somatic mutations in pre-miR-142 (the only example of recurrently mutated miRNA gene in cancer), shown to contribute to the development of diffuse large B-cell lymphoma, follicular lymphoma and acute myeloid leukemia [36,37,38].

The importance of miRNA gene sequences is reflected by their high conservation and low genetic variability [39,40,41]. Previous studies by our group and others have shown that the density of single nucleotide polymorphisms (SNPs) is significantly lower in miRNA genes than in their flanking sequences or in the whole genome [42,43,44,45]. We have also shown that miRNA genes less frequently overlap with common copy number variations (CNVs) [43]. On the other hand, SNPs located in miRNA genes may affect miRNA biogenesis and the specificity of miRNA target recognition. For example, the G>C substitution (SNP rs138166791) in the penultimate position of pre-miR-890, significantly lowers the cleavage efficiency by DROSHA and, consequently, decreases the levels of mature miR-890-5p and miR-890-3p [46]. There is also evidence that some SNPs within miRNA genes may correlate with different diseases, including cancer [47,48,49,50,51,52,53]. Finally, there are a few examples of Mendelian diseases caused by germline mutations in miRNA genes [54,55,56,57].

To determine the frequency and potential consequences of somatic sequence variants in miRNA genes, as well as to identify potential overmutated hotspot miRNA genes, we searched for sequence variants in 1642 miRNA genes using whole exome sequencing (WES) datasets of 1066 samples of the two most common types of lung cancer, i.e., LUAD and LUSC. As a result, we identified over 1000 somatic sequence variants in more than 500 different miRNA genes and showed that a substantial fraction of the mutated miRNA genes overlap in the two types of lung cancer. We characterized all the variants in terms of localization (in subregions of miRNA precursors), type, and potential functional consequences. Among the identified variants were mutations in well-known oncomiRs and suppressormiRs, including let-7, miR-21, and miR-205. A substantial fraction of the identified variants were localized in sequences of mature miRNAs, including in seed sequences and DROSHA and DICER cleavage sites. We performed an analysis with a set of computational tools and showed that somatic mutations in miRNA genes may affect (i) target recognition, (ii) the structure of miRNA precursors, and (iii) structural/sequence motifs that play a role in RNA:protein interactions important for miRNA biogenesis. Finally, we identified and characterized several significantly overmutated hotspot miRNA genes that may be potential driver oncomiRs or suppressormiRs.

## 2. Results

### 2.1. General Characteristics of the Identified Somatic Sequence Variants

To identify somatic sequence variants in miRNA genes, we took advantage of somatic mutation calls performed on WES datasets of 569 LUAD and 497 LUSC paired tumor/normal samples generated within The Cancer Genome Atlas (TCGA) project. We focused our analysis on miRNA genes encoding the most crucial and well-defined sequences of primary miRNA precursors, i.e., pre-miRNA-coding sequences and flanking sequences extending 25 nt upstream and downstream. The sequencing datasets covered 80% of known human miRNA genes (1642 of nearly 2000) (Appendix A).

In the set of selected miRNA genes, we identified 545 and 546 somatic sequence variants in 350 and 353 miRNA genes in LUAD and LUSC, respectively (Figure 1A and Appendix A). More than half of the mutated miRNA genes are annotated as high confidence in miRBase. Among them are genes of miRNAs well known to play an important role in cancer development. An example is the *miR-205* gene one of the most intensively studied cancer-related miRNA acting both as oncomiR and suppresormiR (reviewed in [58]). As shown in Figure 1B, the sets of mutated miRNA genes strongly overlapped between the two types of cancer, and the number of miRNA genes mutated in these two cancers significantly exceeded the number expected by chance (if the occurrence of identified variants in both cancers would be independent; hypergeometric probability with normal approximation test: fold enrichment = 2.3; *p* < 1 × 10^−45^). The distribution of mutation types was similar between the two cancers, with substitutions, indels, and complex sequence changes accounting for ~91%, 4%, and 5%, respectively. At least one variant was found in ~50% and ~60% of LUAD and LUSC samples, respectively (Appendix A). Over 75% of the mutated samples had either one (~49%) or two (28%) sequence variants, and less than 3% had more than 5 variants. To verify the reliability of our data processing method (passing variant thresholds), in the similar way, we performed a mutation analysis in the well-known LUAD protein-coding cancer drivers. The mutation frequencies in the particular drivers were consistent with that determined before (r^2^ = 0.92; *p* = 0.0002) [9].

As shown in Figure 1C, the distribution of somatic sequence variants across chromosomes only roughly correlated with the number of miRNA genes located on a particular chromosome (LUAD: r^2^ = 0.465, *p* = 0.0003; LUSC: r^2^ = 0.443, *p* = 0.0005). Exceptions were overmutation of chromosomes 19 (in both cancer types) and 14 (in LUSC) and undermutation of chromosomes 1 and 2 (in LUSC and LUAD, respectively). Overall, the most abundantly mutated chromosomes were 14, 19, and X, encompassing 20% of the analyzed miRNA genes, but cumulatively accounting for 40% of the identified variants (Figure 1D).

### 2.2. Somatic Mutations Are Not Equally Distributed Along miRNA Precursors

To look closer at the localization of sequence variants in subregions of miRNA precursors, we superimposed the identified variants (substitutions only) on the consensus miRNA precursor structure (Figure 2A) and categorized them according to localization in the miRNA gene subregions (Table 1 and Appendix A). As shown in Figure 2 in both LUAD and LUSC samples, sequence variants were more or less evenly distributed along the miRNA precursor sequence, with slight enrichment in miRNA duplex vs. non-miRNA duplex sequences (5.9 vs. 4.5 mutations/Mbp [mut/Mbp], *p* = 0.0012, and 6.9 vs. 5.5 mut/Mbp, *p* = 0.0078, in LUAD and LUSC, respectively (binomial distribution)). 

A similar sequence variant distribution was observed when precursors of predominantly 5′- and 3′-miRNAs were analyzed separately (Figure 2B,C, lower panels) and when the analysis was narrowed to the precursors of high-confidence miRNAs defined by either miRBase or MiRGeneDB [14,59] (Appendix A). Therefore, the division of the identified variants into the functional subregions of miRNA gene revealed the further differences in variant distribution, with the highest density in the mature miRNA sequence (including seed) and the lowest in the 5′flanking sequence, consistently in LUAD and LUSC (Table 1).

### 2.3. Mutations in miRNA Seed Regions Alter mRNA Target Recognition

It may be expected that the effect of a miRNA gene mutation will strongly depend on its localization. The most unequivocal effect is caused by variants in the miRNA seed region responsible for miRNA:target interactions. Any change in this region may significantly alter the spectrum of regulated transcripts. To obtain deeper insight into the scale of changes in target recognition caused by seed variants, we employed TargetScan to identify potential targets of corresponding wild-type and mutated miRNAs (Figure 3 and Appendix A). The analysis was performed for 48 and 56 seed variants in LUAD and LUSC, respectively. As expected, the target overlap between wild-type and mutated miRNAs was generally low and usually did not exceed a few percentage points. The highest overlap of targets was observed for changes in the last (7th) position of the seed sequence. Additionally, seed changes may cause either a decrease or an increase in a number of predicted targets. As shown for miR-518d-3p, two sequence changes located in adjacent positions of the seed sequence have completely different effects on the number of predicted targets (Figure 3). The extreme examples are the changes at the 2nd position of the miR-1306-3p seed (167× increase in the number of predicted targets, 5 vs. 836) and at the 2nd position of the miR-922-3p seed (48× decrease, 387 vs. 8) (Figure 3 and Appendix A).

### 2.4. Somatic Mutations May Alter Interactions of miRNA Precursors with Regulatory Proteins

An increasing number of proteins regulating miRNA biogenesis have been recently recognized [19]. Disruption of structural or sequence motifs recognized by these proteins may affect the interaction of miRNA precursors with proteins and, consequently, miRNA biogenesis. Therefore, we employed recently developed by one of us (M.O.U.T) miRNAmotif software [60] to investigate whether the identified somatic sequence variants disrupt or create known sequence motifs recognized by the following regulatory proteins: hnRNPA1, HuR, KSRP, Lin28, MBNL1, MCPIP1, DGCR8, MATR3, ZC3H7, YBX1, TRIM71, PTBP1/3, DDX17, RBFOX, SMAD, CELF1/2, and ZC3H10. The analysis led to the identification of 84 mutations disrupting 55 motifs and creating 54 motifs (some mutations affected more than one motif) (Table 2 and Appendix A). In our study, the most frequently affected sequence motifs were UGU and VCAUCH recognized by the DGCR8 and DDX17 regulatory proteins, respectively.

### 2.5. Identification of Hotspot miRNA Genes

Analysis of the sequence variants distribution over the analyzed regions showed that 8 (*miR-890*, *miR-664b*, *miR-1297*, *miR-379*, *miR-1324*, *miR-892a*, *miR-887*, and *miR-509-3*) and 2 (*miR-527* and *miR-592*) hotspot miRNA genes were significantly overmutated in LUAD and LUSC, respectively (Table 3). To further evaluate the reliability of the identified hotspot miRNA genes, we re-calculated mutation enrichment significance, weighting the mutation occurrences by the following factors: 2×, mutations in seeds; 1.5×, mutations in miRNAs (guide-strand only); 1.5×, mutations affecting the functional motifs or DROSHA/DICER cleavage sites; 1×, other mutations. As shown in Table 3 the p-value calculated based on the weighted-mutation values further decreased for 8 hotspot-miRNA-genes, including the *miR-509-3*, *miR-1324* and *miR-379* genes with an excess of mutations affecting the mature miRNA and functional motif sequences. Eight of the 10 hotspot miRNA genes were defined as high confidence in either miRBase or miRGeneDB. As shown in Figure 4, there were no specific hotspot mutations but rather randomly distributed mutations over the sequences of hotspot miRNA genes. An exception may be the *miR-890* and *miR-887* gene, in which most mutations clustered within the DROSHA cleavage site and basal stem junction in the 5′ flanking region, respectively. Analysis of the expression of particular miRNAs showed that miR-887 levels in all mutated samples were below the median of this miRNA in LUAD (Appendix A). A similar observation was made for mutations in the *miR-664b* gene, whereas mutations in other hotspot miRNA genes showed a more or less random distribution over the range of miRNA expression levels.

The co-mutation plot shown in Figure 5A presents the basic clinical and epidemiological characteristics of cancers with sequence variants in miRNA genes (emphasizing mutations in hotspot miRNA genes). The formal analysis (Appendix A), although of low statistical power, showed an association of *miR-890* gene mutations with later tumor stage (LUAD: stage I-II vs. III-IV, *p* = 0.019), more frequent mutations in the *miR-887* gene in males (*p* = 0.045), and a borderline significant correlation of *miR-664b* gene mutations with lower cigarette-per-day-smoking in LUAD. However, due to the low number of the identified variants in particular hotspot miRNA genes, the abovementioned associations (not corrected for multiple comparisons) must be interpreted cautiously and cannot be generalized without further validation.

As shown in Figure 5B, although mutations in hotspot miRNA genes and known driver genes are not mutually exclusive, a substantial fraction of mutations in hotspot miRNA genes occurred in samples with no known driver gene mutations.

### 2.6. Somatic Mutations May Affect miRNA Precursor Structure

Another consequence of mutations in miRNA genes are structural changes that can be caused by mutations in any subregion of the miRNA precursor. Using mutations in the hotspot miRNA genes as examples, we compared the secondary and spatial structures of wild-type and mutant miRNA precursors. The structure prediction was performed with the use of the mfold [104] and RNAComposer [105] software. This analysis showed that most mutations (substitutions) induced subtle, local changes, but some caused more severe deformations of the structure (Figure 6). An example of a mutation that induced serious structural aberrations in the hairpin structure of miRNA precursor is n.57C>A in the 3′ arm of pre-miR-664b (Figure 6A). Such mutations very likely affect miRNA biogenesis (processing by cellular RNases). Mutations that introduced enlarged internal loops, e.g., n.20G>T and n.58T>A in the miR-664b precursor and n.66G>T and n.21G>A in the miR-890 precursor although do not affect substantially the secondary structure of the precursors, may increase the flexibility of the helix axis, manifested by changes in the geometry of 3D structure. 

Changes in the predicted secondary structure, which affect pre-miRNA stability, are not always manifested in spatial geometry, as shown for the n.60T>C mutation in the miR-890 precursor, which caused the loss of two nucleotide internal loops and increased the rigidity of the precursor stem but did not affect the 3D structure of the precursor.

### 2.7. Experimental Identification of Mutations in Hotspot miRNA Genes

In the next step, we used Sanger sequencing to screen a small group of ~80 NSCLC samples for somatic sequence variants in three of the identified hotspot miRNA genes, i.e., the most frequently mutated *miR-890*, and the most frequently annotated with cancer-related functions *miR-379* and *miR-1297* genes (Table 3). In total, we detected 5 sequence variants (Appendix A, Figure 4, and Appendix A). Among the identified sequence variants was n.66G>C in the *miR-890* gene, identified in the hemizygous state in 3 independent cancer samples (Appendix A). The mutation is located in the 3′ arm of the precursor, next to the predicted DROSHA cleavage site, at the same position as a mutation identified in the TCGA cohort (n.66G>T). Additionally, mutations (n.1-18A>G) in the 5′ flanking region of the *miR-890* gene were detected in two other samples. In addition, a single mutation (n.77+26C>T) in 3′ flanking region, 36 nucleotides upstream of the predicted pre-miRNA sequence was detected. Mutations were also found in the *miR-379* gene, one (n.13C>A) in the seed region (Appendix A) and the other (n.1-37C>T) in the 5′ flanking region, 42 nucleotides upstream of the predicted pre-miRNA sequence. As we do not have access to corresponding normal DNA samples, we cannot unambiguously confirm the somatic status of the identified changes.

## 3. Discussion

Cancer development is associated with the accumulation of numerous genetic aberrations, including point mutations, CNVs, and copy number-neutral genomic rearrangements. Most of these alterations are neutral (passenger) changes occurring randomly throughout the genome, but others may be cancer-driving mutations. The accumulation of such mutations in a particular gene or the recurrent occurrence of a particular mutation may indicate its importance in cancer development. To date, many driver genes/mutations have been identified, some of which are utilized as biomarkers in personalized cancer therapy [106]. Prominent examples include *EGFR* mutations in lung cancer and *JAK2* mutations in myeloproliferative disorders [107,108]. So far, most of the attention has been on the analysis of protein-coding genes; therefore, the majority of experimental approaches and bioinformatics algorithms used for somatic mutation detection and identification of potential driver signals are not well suited for non-protein-coding parts of the genome. The limitations of tools dedicated to the analysis of genetic variation in non-coding sections of the genome (~99%) and thus the limited number of studies focusing on these regions were recently described and discussed [34,109]. It should also be noted that a recent increase in interest in non-coding genomic variation, most likely inspired by the identification of highly recurrent mutations in the promoter of the *TERT* gene in melanoma and other cancers [110], is focused mostly on regions playing a role in DNA:protein interactions (e.g., promoters and enhancers) [109].

Considering the role of miRNAs in regulating gene expression, it can be presumed that somatic mutations may affect numerous cellular processes through changes in miRNA functionality. Somatic mutations in miRNA genes may manifest as aberrations in miRNA:target interactions induced predominantly by mutations in the miRNA guide strand sequence, especially but not exclusively in the seed region [111], or as aberrations in miRNA biogenesis caused either by mutations in essential regulatory motifs or mutations affecting miRNA precursor structure. Somatic mutations in miRNA genes may, therefore, cripple or enhance their silencing properties or create new miRNAs, i.e., miRNAs recognizing new targets. Thus, the accumulation of somatically acquired gain- or loss-of-function mutations in a specific miRNA gene may benefit cancer development and progression.

In this study, we detected 1091 somatic sequence variants in 521 miRNA genes (Figure 1A) and showed that approximately 50% of the analyzed cancers had at least one such sequence variant in at least one miRNA gene. It has to be noted, however, that similarly, as in protein-coding genes, the great majority of sequence variants detected in miRNA genes constitute spontaneous mutations accumulating during cancer development and only a small fraction of these variants may be a potential driver or functional mutations. Our results show that somatic sequence variants are generally spread throughout the genome, even though some chromosomes were overmutated (Chr14 and Chr19) or undermutated (Chr1 and Chr2). Such differences in the mutation load cannot be explained by overall differences of mutation frequency in different chromosomes and were not observed for protein-coding regions [112,113,114]. As some excess of miRNA-gene mutations at ChrX occurs both in man and woman, it is unlikely to be caused by the observed before hypermutation of the inactive ChrX [112]. To better understand the potential function of the identified variants, we characterized them with respect to localization in different subregions of miRNA precursors, influence on miRNA precursor secondary and tertiary structures, impact on sequence motifs bound by different proteins that participate in miRNA maturation and changes in mature miRNA:target interactions.

Our analysis identified well-recognized suppressormiRs or oncomiRs among the miRNA genes with the identified variants. Examples of the mutated suppressormiR genes may be the *Let-7a-2*, *let-7c,* and *let-7e* genes belonging to the let-7 family, known to have tumor suppressor function by repressing cell proliferation and regulating the cell cycle [115] and the *miR-205* gene, which may act either as suppresormiR inhibiting cancer progression by suppressing EMT or oncomiR accelerating cancer development by facilitating tumor initiation and proliferation [58,116]. It was shown that miR-205 [24,25] belongs to the group of miRNAs most commonly overexpressed and that its gene is frequently gained or amplified in lung cancer [33]. The same n.35C>T mutation in the miR-205 seed sequence was observed in two samples in our studies (Figure 3 and Appendix A). As shown in Figure 3 the mutation causes the loss of many miR-205 targets, including well-validated *MED1*, *ERBB3*, *VEGFA*, *E2F1*, and *PTEN*, that are crucial for the cancer-related functions of miR-205. Less effective downregulation of the tumor suppressor *PTEN* may result in tumor suppression whereas less effective downregulation of such oncogenes as *VEGFA* (the key mediator of angiogenesis) or *E2F1* (the transcription factor controlling cell cycle) may enhance tumor progression, invasion, and/or metastasis. Mutations were also found in genes encoding known oncomiRs, such as the *miR-21*, *miR-155* genes and in the genes (*miR-17*, *miR-18a*, *miR-20a, R19b-1*, and *miR-92a-1*) belonging to the *miR-17-92* cluster, also known as OncomiR-1 [117] (Appendix A). Finally, mutations were also detected in the *miR-143* gene, which encodes the most abundantly expressed miRNA (accounting for ~30% of all expressed miRNAs) and, next to miR-21, contribute to the greatest changes (decreases) in miRNA levels in most cancer types [118].

Among the mutated miRNA genes, we identified a group of ten hotspot miRNA genes that were significantly overmutated in either LUAD or LUSC samples. The fact that 80% of hotspot miRNA genes (64% of all mutated miRNA genes) but only 45% of all analyzed miRNA genes were assigned as high confidence according to miRBase and/or miRGeneDB supports the reliability of our findings. The dispersed distribution of mutations over the hotspot miRNA genes (without specific hotspot mutations) is consistent with the pattern of loss-of-function mutations typically observed in suppressor genes [119]. This finding coincides with previous results (summarized in Table 3) suggesting that most hotspot miRNAs act as suppressormiRs. An example is miR-379, which downregulates MDM2, thus preventing the ubiquitination and degradation of p53 [120]; PDK1, which is the convergence point of cancer signaling pathways such as the PI3K/Akt, Ras/MAPK and Myc pathways [121]; and COX-2, which inhibits the transition from acute to chronic inflammation and thereby prevents cancer initiation and progression [122]. An exceptional mutation distribution pattern was the accumulation of mutations in a basal region of the *miR-887* gene and the introduction of a basal UG motif, which may be associated with the oncogene-like pattern of mutations resulting in accelerated pre-miRNA processing and thus an elevated level of this miRNA in cells. Increased levels of miR-887 were previously reported in endometrial and rectal cancer [86,87]. In contrast, other authors have demonstrated that miR-887 may act as a tumor suppressor in prostate cancer, breast cancer and small cell lung cancer (SCLC) [88,89,90]. We have shown that mutations in miRNA genes and known cancer driver genes are not mutually exclusive, but a substantial number of miRNA gene mutations were identified in samples without mutations in known protein-coding drivers. We also showed that mutations in some hotspot miRNA genes correlated with different epidemiological and clinical characteristics of cancer and that *miR-887* and *miR-664b* gene mutations were associated with lower expression of these specific miRNAs. Nevertheless, the statistical power of these analyses was low (low number of mutations in a particular gene), and therefore, the determination of whether the identified hotspot miRNA genes have cancer-driver potential requires further statistical and experimental validation. On the other hand, even if the detected somatic sequence variants are not primary drivers, they may still play a role in cancer. It was recently shown that passenger mutations may both prevent further cancer progression [123] and cooperate in driving cancer development and drug resistance [124,125].

Somatic mutations in the miRNA duplex lead to altered miRNA:target interactions (Figure 3 and Appendix A) and also may affect miRNA maturation by changing the precursor structure (Figure 6). For example, the n.66G>T mutation in the 3′ arm of the miR-890 precursor destabilizes the duplex structure and enlarges the internal loop at the predicted DROSHA cleavage site. This mutation may affect DROSHA cleavage or change its specificity. It was shown before that the G>C substitution (SNP rs138166791) located at the same position significantly decreases the effectiveness of pre-miRNA processing and the level of mature miRNA [46]. Additionally, as we have previously shown, such nucleotide substitutions may alter DROSHA cleavage specificity and may change the pool of generated isomiRs [126,127,128]. Interestingly, although the minor allele of this SNP is very rare (<1% in the GnomAD, TOPMED, ExAC, and 1000G databases), we detected this allele as a hemizygous variant in 3 of ~60 lung cancer samples in our experimental analysis.

We have also shown that somatic mutations in miRNA genes may affect (destroy or create) sequence motifs recognized by proteins playing a role in miRNA maturation. The most frequently gained sequence motifs were the VCAUCH and UGU motifs recognized by DDX17 and DGCR8, respectively. Both motifs are important for proper pri-miRNA recognition and processing by DROSHA. Another frequently created motif was the UGC motif that is bound by MCPIP1, which antagonizes DICER and terminates miRNA biosynthesis via cleavage of the pre-miRNA terminal loop and via further precursor degradation [129]. The DGCR8-binding motif was also the most frequently lost due to somatic mutation, followed by the CAGAC motif, which is recognized by SMAD and enhances pri-miRNA processing by the microprocessor complex. Additionally, TGF-β-stimulated SMAD may enhance the transcription of miRNA-coding genes by binding to their promoters [130,131]. As shown in Figure 4, mutations may also affect the SRSF3-binding CNNC motif involved in pri-miRNA processing [132].

In contrast to the numerous approaches/algorithms for identifying driver genes/mutations in protein-coding genes (e.g., MutSig2CV, HotSpot 3D, CLUMPS, PARADIGM, HotNet2, e-Driver and 20/20+) [133,134]), which take advantage of predictable consequences of mutations in affected proteins, the identification of drivers in non-protein-coding regions is limited to the analysis of mutation frequency/distribution in a region of interest. The challenge in identifying driver mutations in non-coding regions stems mostly from the lack of a simple code (such as the protein code) that would allow one to predict the function of mutations and to distinguish deleterious from benign or neutral mutations. Approaches such as MutSigNC and LARVA, which utilize analysis of background mutation distribution, were recently modified for the identification of non-coding drivers, but they remain mostly limited to regions such as promoters, enhancers, and transcription factor binding sites [109]. To the best of our knowledge, there is currently no tool dedicated to the automated and statistically supported identification of driver mutations in miRNA genes. Therefore, our results, as well as the recently published tool ADmiRE [34], may help to prioritize potentially functional mutations and to develop a better algorithm for the identification of driver mutations in regions coding for non-coding RNAs, particularly miRNAs. For example, an analysis of mutations in hotspot miRNA genes detected in this study showed that excess mutations in the miR-509-3 gene occur in the mature miRNA sequences and that those in the miR-887 gene affect regulatory sequence motifs.

## 4. Materials and Methods

### 4.1. Data Resources

We used molecular and clinical data (Level 2) for LUAD and LUSC generated and deposited in the TCGA repository (http://cancergenome.nih.gov). These data included results of somatic mutation calls in WES datasets of 569 LUAD and 497 LUSC samples analyzed against matched normal (non-cancer) samples with the use of the standard TCGA pipeline. The normal (non-cancer) DNA samples were extracted either from peripheral blood samples, adjacent histologically-confirmed normal tissue samples resected at surgery, or from both. The used paired tumor/normal approach permitted the identification and elimination of germline variants. We analyzed annotated somatic mutations with corresponding clinical information and miRNA expression data.

### 4.2. Data Processing

We analyzed somatic sequence variants in 1642 miRNA gene regions (Appendix A) including 1600 regions covered by the probes used for exome enrichment (SureSelect, Agilent, Santa Clara, CA, USA) and 42 regions co-captured during library enrichment but not directly covered by the SureSelect probes. The miRNA genes cover extended pre-miRNA-coding sequences, including 25 nucleotides upstream and downstream of the pre-miRNA-coding sequence. Pre-miRNA-coding sequences were reconstructed based on 5′ and 3′ mature miRNA coordinates annotated in miRBase v.21 (in cases when only one miRNA strand was indicated in miRBase, the other pre-miRNA end was predicted assuming the pre-miRNA hairpin structure with a 2-nt 3′ overhang). According to the number of reads generated from the particular pre-miRNA arm (miRBase), the analyzed precursors were classified into one of 3 categories: (i) pre-miRNAs generating mature miRNA predominantly from the 5′ arm (≥90% of reads from the 5′ arm); (ii) pre-miRNAs generating mature miRNA predominantly from the 3′ arm (≥90% of reads from the 3′ arm); and balanced pre-miRNAs (>10% of reads from each arm). To validate the accuracy of the identified miRNA gene variants (assumed threshold/filters), we also analyzed the occurrence of somatic point mutations in selected hotspot exons of well-known NSCLC driver genes [9], i.e., *KRAS*, *EGFR*, *BRAF*, *HRAS*, *NRAS*, *MAP2K1*, *MET*, *ERBB2*, *RIT1*, and *NF1* (Appendix A).

Some analyses (indicated in the text) were performed on a narrowed list of high-confidence miRNA genes defined either in miRBase v.22 (*n* = 616) or MiRGeneDB (*n* = 521). The precursors deposited in MiRGeneDB are defined based on criteria that include careful annotation of the mature versus passenger miRNA strands and evaluation of evolutionary hierarchy; therefore, they are much more conservative than those in miRBase [59].

From the WES data generated with the use of four different algorithms (MuSE, MuTect2, VarScan2, and SomaticSniper), we extracted somatic mutation calls with PASS annotation. The extraction was performed with a set of in-house Python scripts available at (https://github.com/martynaut/mirnaome_somatic_mutations). To avoid duplicating variants detected in multiple datasets associated with the same patient, we combined files summing reads associated with particular sequence variants. The lists of somatic sequence variants detected by different algorithms were merged such that variants detected by more than one algorithm were not multiplicated. To further increase the reliability of the identified somatic sequence variants (and avoid the identification of uncertain variants), we removed those that did not fulfill the following criteria: (i) at least two alternative allele-supporting reads in a tumor sample (if no alternative allele-supporting read was detected in the corresponding normal sample); (ii) at least 5x higher frequency of alternative allele-supporting reads in the tumor sample than in the corresponding normal sample; (iii) somatic score parameter (SSC) > 30 (for VarScan2 and SomaticSniper); and (iv) base quality (BQ) parameter for alternative allele-supporting reads in the tumor sample > 20 (for MuSE and MuTect2). These additional criteria resulted in the exclusion of 193 sequence variants in miRNA genes and 93 mutations in cancer driver genes. Finally, the average sequencing depth of the identified somatic sequence variants in miRNA genes was ~290 for LUAD and ~313 for LUSC. In 15 LUAD and 10 LUSC samples, two somatic alterations were detected within one miRNA gene. Although we could not distinguish whether such changes occurred on one or two different alleles, for simplicity, we treated each of these pairs of mutations as one complex mutation. All sequence variants were designated according to HGVS nomenclature in reference to coordinates of corresponding miRNA precursors specified in miRBase v.21.

### 4.3. Statistics

Unless stated otherwise, all statistical analyses were performed with statistical functions in the Python module scipy.stats. Particular statistical tests are indicated in the text, and *p* < 0.05 was considered significant. If necessary, p-values were corrected for multiple tests with the Benjamini-Hochberg procedure [135]. Hotspot miRNA genes were identified based on the probability of occurrence of the observed number of mutations, which was calculated with the use of 2-tailed binomial distribution (VassarStats: web site for statistical computation http://vassarstats.net/), assuming a background random occurrence of identified mutations in all analyzed miRNA genes and considering the miRNA gene length.

### 4.4. Target, Structure, and miRNA Motif predictions

Target predictions were performed with the TargetScan Custom (release 5.2) on-line tool [136], and the results are presented as area-proportional Venn diagrams [137].

The secondary structures of miRNA precursors were predicted using mfold software with default parameters. Three-dimensional structural predictions of pre-miRNA stem-loop structures were generated using RNAComposer software with default parameters [105] and visualized in PyMOL (Schrödinger, LLC, New York, NY, USA).

To identify mutations affecting sequence motifs recognized by pre-miRNA-interacting proteins, we modified miRNAmotif software (https://github.com/martynaut/mirnamotif, [60]), which is dedicated to the identification of sequence motifs in wild-type pre-miRNA sequences. As miRNAmotif utilizes coordinates of pre-miRNA sequences deposited in miRBase, we did not analyze the effect of mutations in sequence outside the coordinates, and we analyzed only substitutions. For proteins known to interact with loop structures of pre-miRNAs, we performed a search in the loop sequence only (in miRNAmotif this sequence is referred to as the linking sequence). The motif search was performed in the 5′-3′ direction.

### 4.5. Lung Sample Screening

For the experimental identification of sequence variants in the *miR-890*, *miR-379*, and *miR-1297* genes, we analyzed DNA from 84 tumor samples (formalin-fixed paraffin-embedded) with histopathologically confirmed NSCLC diagnosed at the Franciszek Lukaszczyk Oncology Center in Bydgoszcz (central Poland). The study was approved by the Committee of Ethics of Scientific Research of Collegium Medicum of Nicolaus Copernicus University, Poland (KB 42/2018). The data were analyzed anonymously. The appropriate genomic fragments were amplified by PCR using the following primers: *miR-890* gene: F-5′GAACAAGCTCGTTTTCTGTTCTT3′, R-5′CAGTGGGCTGGAAATTCTCT3′ (444 bp product, annealing temperature 60 °C); *miR-379* gene F-5′CAAATCCAGCCTCAGAAAGC3′, R-5′TGGAGCAGTGCTGAAGCTAA3′ (246 bp, 60 °C); and *miR-1297* gene F-5′TCAAGGGTGATAAGAAAGAGGA3′, R-5′GATTTTCATAGGACAACATCTTCA3′ (250 bp, 58 °C). PCR was performed according to the manufacturer’s recommendations (GoTaq DNA polymerase protocol, Promega, Madison, WI, USA). All PCR products were purified using the EPPIC Fast kit (A&A Biotechnology, Gdynia, Poland) and sequenced directly using the BigDye v3.1 kit (Applied Biosystems, Foster City, CA, USA) and an ABI PRISM 3130xl genetic analyzer (Applied Biosystems, Foster City, CA, USA). High-quality PCR products of the *miR-890*, *miR-379*, and *miR-1297* genes were obtained and sequenced from 59, 71, and 75 samples, respectively. All detected sequence variants were confirmed by sequencing in two directions.

## 5. Conclusions

In summary, in this study involving WES data from human lung tumors, we revealed a plethora of somatic sequence variants within miRNA genes and addressed their potential functional consequences. Although our results must be further evaluated and experimentally validated, they provide a good starting point for discussion and further research on the development of miRNA gene-dedicated computational approaches, which may help elucidate the role of somatic miRNA gene mutations in cancer in the future. Our findings may be helpful in cracking the code of so-called non-coding DNA and may contribute to fully understanding the role of miRNAs in cancer development and to examining many yet unexplored parts of the genome.

## Figures and Tables

**Figure 1 cancers-11-00793-f001:**
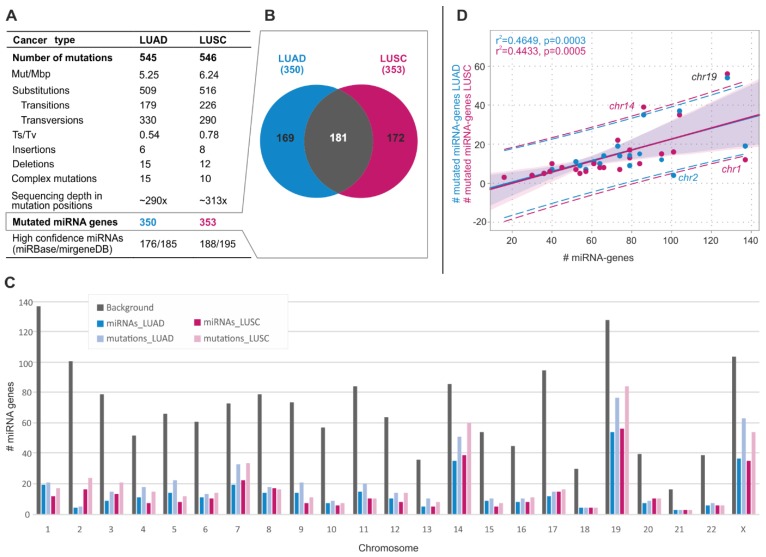
General characteristics of somatic sequence variants in miRNA genes in LUAD and LUSC. (**A**) Table summarizing the main statistics and type of identified variants in LUAD and LUSC. (**B**) Venn diagram showing the overlap of miRNA genes mutated in LUAD and LUSC. (**C**) Distribution of miRNA gene variants over chromosomes. Gray bars show the number of miRNA genes analyzed in the study (background). Blue and pink bars show the number of mutated miRNA genes in LUAD and LUSC, respectively. Light blue and light pink bars show the number of variants in miRNA genes in LUAD and LUSC, respectively. (**D**) Correlation of the total number of miRNA genes (x-axis) and the number of mutated miRNA genes (y-axis) on a particular chromosome (dots). Blue and pink dots and regression lines indicate the results for LUAD and LUSC, respectively. Corresponding r^2^ and p-values are indicated on the graph. Dashed lines represent the confidence interval with 95% probability.

**Figure 2 cancers-11-00793-f002:**
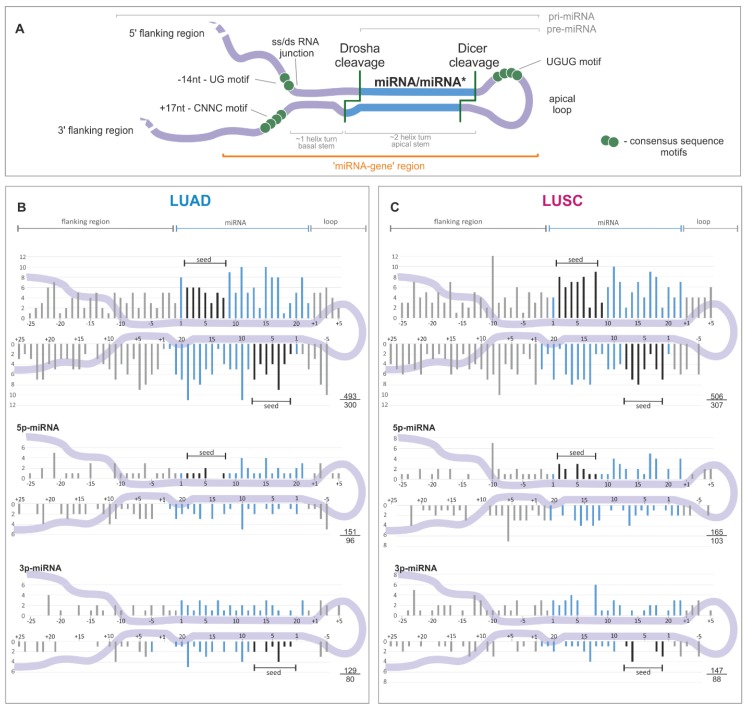
Distribution of somatic sequence variants within miRNA precursors. (**A**) An overview of a primary miRNA transcript with indicated subregions considered in the study. The miRNA duplex is indicated in blue, and sequence consensus motifs recognized as enhancers of miRNA biogenesis are represented by green circles. (**B**) and (**C**) The distribution of substitutions in the subregions of miRNA precursors for LUAD and LUSC samples, respectively. miRNA duplex positions are indicated in blue, seed regions in black, and flanking regions and terminal positions of the apical loop in gray. The numbers in the lower-right corner represent the number of plotted substitutions (upper) and the number of mutated miRNA genes (lower). Analyses were also performed in narrowed groups of miRNAs that preferentially release the guide miRNA strand from the 5′ or 3′ arm (below). If present, sequence variants localized beyond position 22 in miRNA are shown cumulatively at position 22. As the size and structure of loops differ substantially among miRNA precursors, the mutation density maps do not show variants located inside the loops.

**Figure 3 cancers-11-00793-f003:**
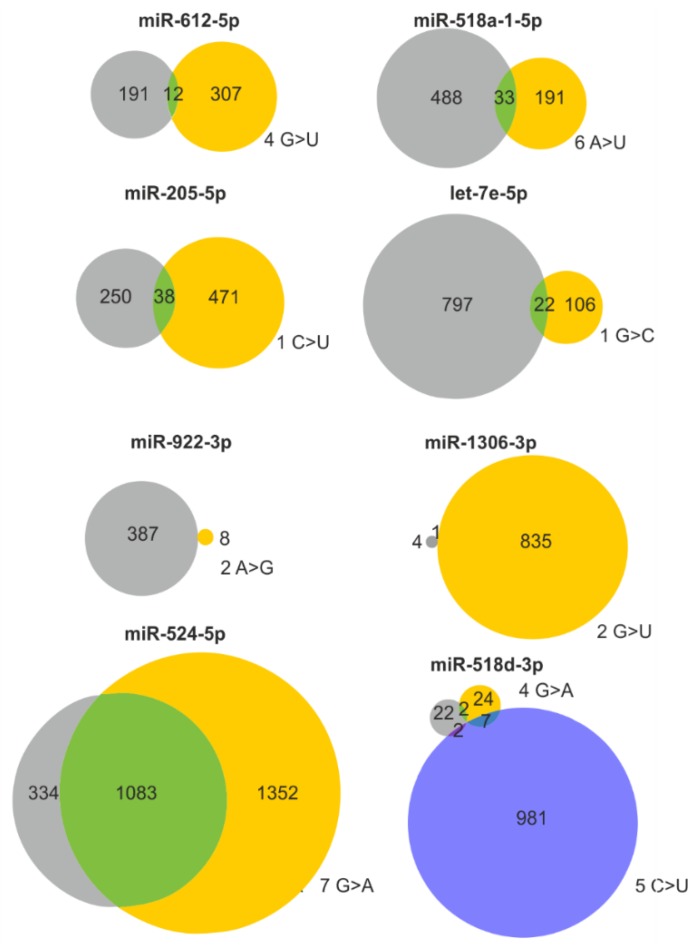
Consequences of selected seed mutations on a pool of predicted target genes. Venn diagrams showing the effect of representative seed mutations on target recognition. Gray and yellow circles indicate predicted targets of the wild-type and mutant seeds, respectively. If more than one sequence variant occurs in a particular seed, the effect of the second variant is shown as a blue circle. The position in the seed sequence and the nucleotide change are shown next to the corresponding circles.

**Figure 4 cancers-11-00793-f004:**
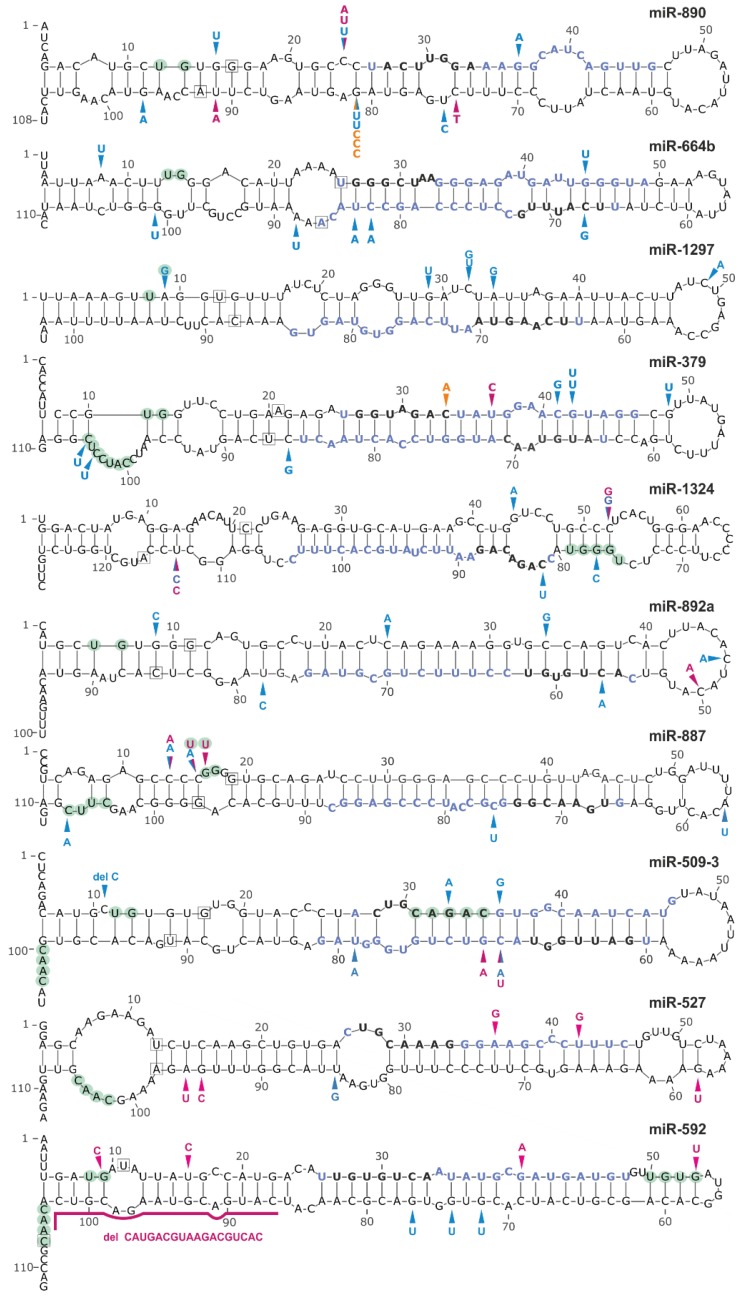
Localization of mutations in hotspot miRNA genes. Schematic secondary structure representations (generated with mfold) of miRNA precursors are shown. Blue, pink, and orange arrowheads indicate sequence variants detected in the LUAD and LUSC datasets and in the panel of lung cancer samples analyzed experimentally in this study, respectively. Light green circles indicate nucleotide positions within sequence consensus motifs or motifs bound by regulatory proteins. Blue and black bolded fonts indicate mature miRNA and seed sequences, respectively. The first and last nucleotides of miRNA precursors annotated in miRBase are indicated in squares.

**Figure 5 cancers-11-00793-f005:**
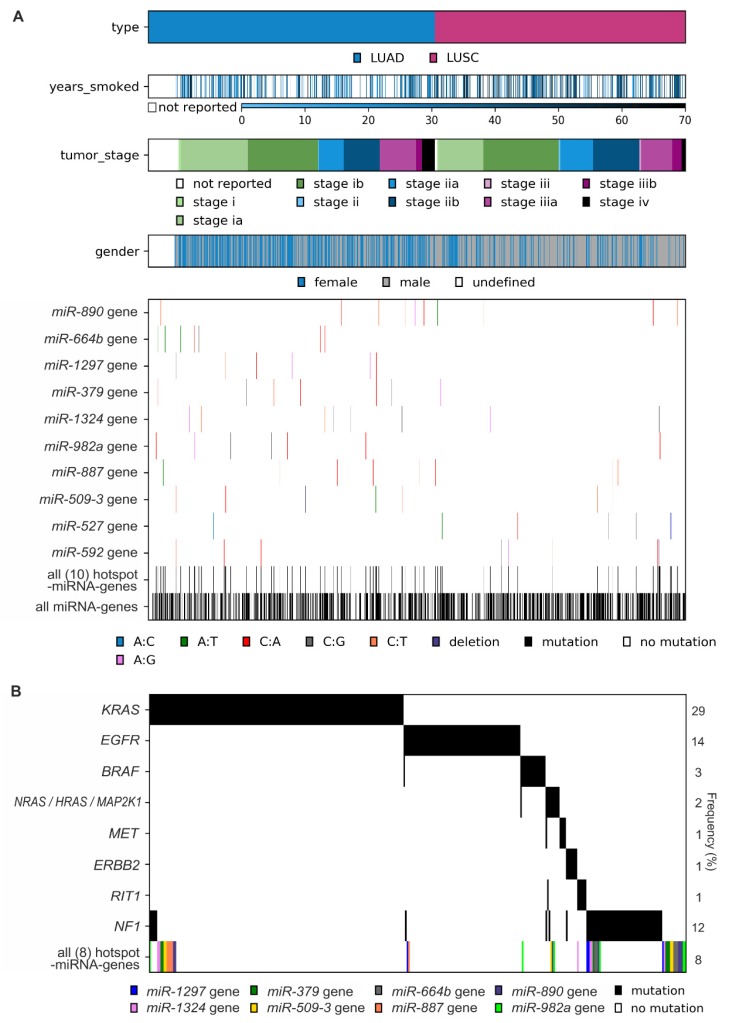
Characteristics of mutations in hotspot miRNA genes. (**A**) Co-mutation plot showing the occurrence of mutations in particular hotspot miRNA genes, all 10 hotspot miRNA genes, and all miRNA genes in cancer samples sorted by cancer type and other cancer characteristics (years smoked, tumor stage, and gender). (**B**) Co-occurrence of mutations in LUAD samples with mutations in known LUAD driver genes and mutations in LUAD hotspot miRNA genes (all 8 hotspot miRNA genes). Note that only 340 (of 569) LUAD samples with mutations in either known cancer drivers and/or miRNA hotspot genes are shown. The frequency of variants in particular genes is shown on the right.

**Figure 6 cancers-11-00793-f006:**
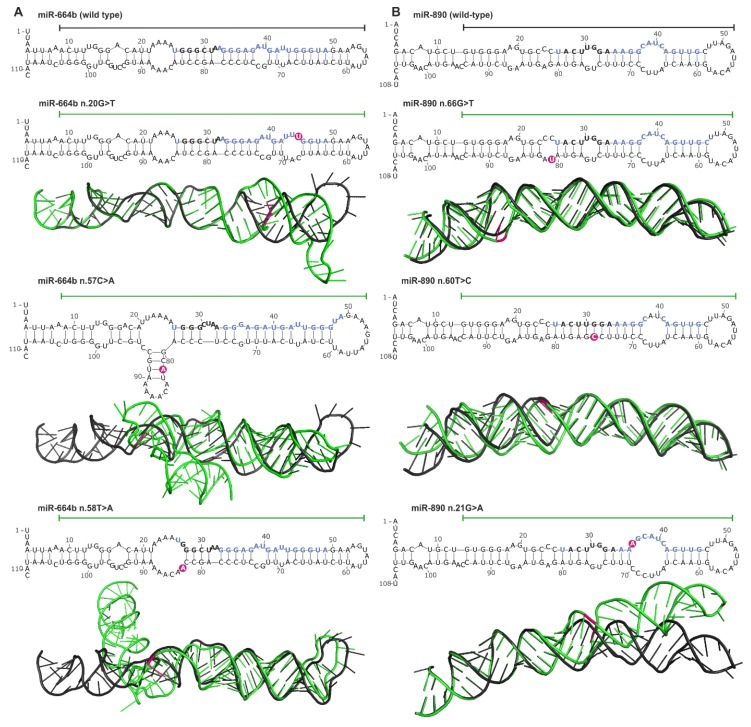
Impact of miRNA gene mutations on the structure of miRNA precursors. (**A**) and (**B**) Representative examples of *miR-664b* and *miR-890* gene mutations, respectively. The effect of each mutation is shown at the levels of secondary and 3D structures. The secondary structures encompass the pre-miRNA precursor sequence and the flanking 25-nt 5′ and 3′ sequences (blue and black bolded fonts indicate miRNA and seed sequences, respectively; pink circles indicate mutation positions). 3D structures encompass sequences indicated by horizontal lines over the corresponding secondary structures. 3D mutant structures (green) are aligned with the corresponding wild-type structures (black). Mutation positions are indicated in pink.

**Table 1 cancers-11-00793-t001:** Distribution of somatic mutations (substitutions) within miRNA precursors.

	LUAD				LUSC			
	N Mutations	Mut/Mbp	Fold Change	*p*-Value	N Mutations	Mut/Mbp	Fold Change	*p*-Value
total	509	4.90	1	-	516	5.90	1	-
5′-flanking region	84	3.60	0.73	0.0010341	103	5.24	0.89	0.1845670
3′-flanking region	104	4.45	0.91	0.2909070	114	5.80	0.98	0.8810662
loop	88	5.23	1.07	0.5342181	70	4.94	0.84	0.1142622
passenger strand	68	4.30	0.88	0.2699724	74	5.56	0.94	0.6339850
guide strand	170	6.92	1.41	0.0000008	155	7.50	1.27	0.0010096
seed region	47	5.89	1.20	0.2197888	53	7.90	1.34	0.0392875

**Table 2 cancers-11-00793-t002:** Effect of somatic mutations on protein binding motifs.

Binding Protein	Sequence Motif	Lost	Gained
hnRNPA1	UAGGGAW	0	0
HuR	AUUUUUAUUUU	0	0
KSRP	GGGU	5	0
Lin28	GGAG	1	3
MBNL1	YGCY	4	5
MCPIP1	UGC	5	8
DGCR8	UGU	14	9
MATR3	AUCUU	1	1
ZC3H7	SMUANY	1	2
YBX1	CAUC	3	5
TRIM71	UAUAA	0	1
PTBP1/3	UUUUUCCNUCUUU	0	0
DDX17	VCAUCH	5	10
RBFOX	GCAUG	2	3
SMAD	CAGAC	8	3
CELF1/2	UGUNNNNNNNUGU	5	3
ZC3H10	GCAGCGC	1	1
Total		55	54

Motifs localized only in the apical loop: hnRNPA1, HuR, KSRP, Lin28, MBNL1, MCPIP1, DGCR8, MATR3, ZC3H7, YBX1, TRIM71, and PTBP1/3.

**Table 3 cancers-11-00793-t003:** Hotspot miRNA genes in LUAD and LUSC.

	miRNA	Cancer-Related Functions	miRTarBase /Literature Validated Targets	miRBase/MiRGeneDB Confidence	Number of Mutations LUAD/LUSC *	*p*-Value Nominal (BH Corrected **)	Weighted Mut. Score LUAD/LUSC *	Weighted *p*-Value Nominal (BH Corrected **)
**LUAD**	**miR-890** chrX (−)	**oncomiR**; inhibits DNA repair and damage response genes; potent IR sensitizing agent due to targeting the *MAD2L2*, *WEE1* and *XPC* genes, downregulated in PRC [61,62]	*MAD2L2*, *WEE1*, *XPC*	−/+	**7**/4	9.2 × 10^-8^ (9.5 × 10^-5^)	**9**/4.5	1.3 × 10^−9^ (1.5 × 10^−6^)
	***miR-664b*** chrX (+)	**suppressormiR**; inhibits proliferation, migration, and invasion and increases the chemosensitivity of *BRCA1*-mutated TNBC cells by targeting *CCNE2* [63]	*CCNE2*	+/−	**7**/0	1.2 × 10^-7^ (9.5 × 10^-5^)	**9.5**/0	1.8 × 10^−9^ (1.5 × 10^−6^)
	***miR-1297*** chr13 (−)	**oncomiR**; promotes cell progression in BC and LUSC by targeting *PTEN,* resulting in activation of *PTEN/PI3K/AKT* signaling pathway [64,65]; **suppressormiR**; suppresses cell proliferation by targeting *TRIB2* and further increasing C/EBPα expression in LUAD [65,66,67]; inhibits the growth and metastasis of CRC by suppressing *CCND2* [68]; suppresses *PTEN* expression and inhibits cell progression in LUSC [65]; promotes apoptosis and inhibits the proliferation and invasion of HCC cells by targeting *HMGA2* [69]; inhibits the growth, migration and invasion of CRC cells by targeting *Cox-2* [66]; suppresses PAC cell proliferation and metastasis by targeting *MTDH* [70]	*TRIB2*, *MALAT1*, *HMGA1*, *HMGA2*, *PTEN*, *CCND2*, *COX2*, *MTDH*	−/−	**6**/0	1.8 × 10^-6^ (7.0 × 10^-4^)	**8**/0	2.4 × 10^−8^ (1.3 × 10^−5^)
	***miR-379*** chr14 (+)	**suppressormiR**; inhibits tumor invasion and metastasis by targeting *FAK* and suppressing *FAK/AKT* signaling in HCC and GC [71,72]; inhibits cell proliferation and invasion in glioma by targeting *MTDH* and inhibiting the *PTEN/AKT* pathway [73]; inhibits BLC growth and metastasis by targeting *MDM2* [74]; inhibits cell proliferation by suppressing *CCND1*; downregulated in BC [75]; inhibits TGF-β-induced IL-11 production in bone metastatic BC cells [76]; suppresses *EIF4G2* and potentiates cisplatin chemosensitivity in NSCLC [77]; suppresses osteosarcoma progression by targeting *PDK1* [78]; inhibits EMT and bone metastasis of PRC [79]; suppresses BC progression by targeting *COX2*; potential therapeutic agent in BC treatment [80]; miR-379/miR-656 cluster is downregulated in multiple human cancers, especially GBM, KIRC and BC [81]; sponged by circHIPK3 [82]	*IL11*, *PTK2*, *CCND1*, *MTDH*, *EIF4G2*, *PDK1*, *COX2*, *MDM2*	+/+	**6**/1	2.4 × 10^-6^ (7.3 × 10^-4^)	**8**/1.5	3.7 × 10^−8^ (1.5 × 10^−5^)
	***miR-1324*** chr3 (+)	**suppressormiR**; targets *FZD5* and downregulates the Wnt/β-catenin signaling pathway in HCC; inhibited by circ_0067934; the circ_0067934/miR-1324/FZD5/Wnt/β-catenin signaling pathway axis is involved in the regulation of HCC progression [83]	*FZD5*	−/−	**6**/2	6.1 × 10^-6^ (1.4 × 10^-3^)	**8**/2	1.2 × 10^−7^ (4.0 × 10^−5^)
	***miR-892a*** chrX (−)	**oncomiR**; upregulated in human CRC tissues and cell lines; promotes cell proliferation and colony formation by suppressing *PPP2R2A* expression [84]; promotes HCC cell proliferation and invasion through targeting *CD226* [85]	*PPP2R2A*, *CD226*	+/+	**6**/1	1.4 × 10^-6^ (7.0 × 10^-4^)	**7.5**/1	3.7 × 10^−7^ (9.8 × 10^−5^)
	***miR-887*** chr5 (+)	**oncomiR**; potential biomarker for ENC; serum miR-887-5p levels are increased in patients with ENC [86]; increased levels in CRC [87]; **suppressormiR**; downregulates *MDM4* in PRC and SCLC by having a high affinity for the *MDM4* rs4245739 SNP C-allele [88,89]; inhibits the invasion of BC cells by targeting *PLD2* [90]	*PLD2*, *MDM4*	+/+	**5**/3	4.6 × 10^-5^ (8.0 × 10^-3^)	**5.5**/3	1.6 × 10^−4^ (2.9 × 10^−2^)
	***miR-509-3*** chrX (−)	**suppressormiR**; decreases cell proliferation, migration and invasion of NSCLC cells by targeting *FOXM1* [91]; attenuates cellular migration and multicellular spheroid formation in OVC by targeting *YAP1* [92]; inhibits the migration and proliferation of GC cells by targeting *XI*AP and of KIRC by targeting *MAP3K8* [93,94]; represses *PLK1,* causing mitotic aberration and growth arrest of LUAD A549 cell line [95]; inhibits the invasion and metastasis of GC by targeting *PODXL* [96]	*NTRK3*, *CFTR*, *XIAP*, *MAP3K8*, *FOXM1*, *YAP1*, *PLK1*, *PODXL*	+/+	**5**/2	4.0 × 10^-5^ 8.0 × 10^-3^)	**5.5**/3	5.8 × 10^−7^ (1.3 × 10^-4^)
**LUSC**	***miR-527*** chr19 (+)	**suppressormiR**; reduces metastatic dissemination by repressing *SMAD4* in osteosarcoma [97]	*SMAD4*	−/+	1/**6**	2.7 × 10^-6^ (7.3 × 10^-4^)	1.5/**7.5**	8.0 × 10^−7^ (1.6 × 10^−4^)
	***miR-592*** chr7 (−)	**oncomiR**; represses *FOXO3* expression and promotes the proliferation of PRC cells [98,99]; upregulation correlates with tumor progression and poor prognosis for patients with CRC [100] **suppressormiR**; targets the *DEK* oncogene and suppresses cell growth in the HCC cell line HepG2 [101]; tumor suppressor in NSCLC by targeting *SOX9* [102]; inhibits cell proliferation, colony formation, migration and invasion in BC by targeting TGFβ-2 [103]	*FOXO3*, *DEK*, *SOX9*	−/+	3/**5**	5.0 × 10^-5^ (8.0 × 10^-3^)	4.5/**5.5**	1.8 × 10^−4^ (2.9 × 10^−2^)

(*) Note that the *p*-value was calculated only based on sequence variants identified in a particular cancer, indicated in bold. (**) *p*-value corrected for multiple comparisons with the Benjamini-Hochberg procedure. BC, breast cancer; BLC, bladder cancer; CRC, colorectal cancer; ENC, endometrial cancer; GBM, glioblastoma multiforme; GC, gastric cancer; HCC, hepatocellular carcinoma; IR, ionizing radiation; KIRC, kidney clear cell renal carcinoma; OVC, ovarian cancer; PAC, pancreatic cancer; PRC, prostate cancer; TGF, transforming growth factor; TNBC, triple-negative breast cancer.

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
