# Peer review of "Somatic Mutations in miRNA Genes in Lung Cancer—Potential Functional Consequences of Non-Coding Sequence Variants"

_cancers, 2019, doi:10.3390/cancers11060793_

Round 1
Reviewer 1 Report
The study by Galka-Marciniak et al. describes the extent of putative somatic mutations of miRNAs genes in lung cancer (lung adenocarcinoma and lung squamous cell carcinoma). The authors identify that mutations target different functional subregions of miRNAs, and that mature miRNA sequences are more mutated than other regions. These mutations may have a functional impact on the predicted interactions with miRNA targets.
They further experimentally confirm the mutations in hotspot miRNA genes by using an independent cohort. Altogether, this study is sound, timely and could lead to the identification of new diagnostic markers and/or therapeutic targets in lung cancers. Of note, the manuscript is well structured and written. In general, the data are cautiously interpreted. Although the main results are convincing, the study could be improved by addressing these specific comments:
- In the first part (page 4 and 5), the authors report a tremendous proportion of miRNAs potentially mutated (500 miRNA genes on 1642 miRNA gene regions analyzed). The material and methods section does not clearly indicate if blood samples of the same patients have been analyzed to distinguish single nucleotide polymorphisms from somatic mutation events. Although interesting, the comparison of mutation frequencies in protein-coding genes may not be sufficient to indicate that any detected substitution is a somatic mutation of a noncoding gene. The authors need to address this issue or minor their finding that 500 miRNAs contain somatic mutations.
- Figure 1D: It would be interesting to perform the same analysis with somatic mutations of protein-coding genes to see if the distribution of mutations in chromosome 1, 2, 14, 19 is a specific feature of miRNAs or linked to lung cancer.
- Figure 3: The functional impact of miRNA mutation through target gene regulation may result in changes in signalling pathways. This can be addressed by performing pathway enrichment analyses of the predicted target genes in both conditions (wt and mutated miRNAs). If specific pathways are enriched, their relevance in lung cancer biology may be discussed. This will reinforce the idea that somatic mutation in miRNAs may have a functional biological impact in lung cancer.
Minor comments:
Figure 2 and 4: another colour code could be used to better highlight the differences in duplex positions and seed regions.
Data presented page 15 on the experimental validation of hotspot miRNA mutations by Sanger sequencing could be moved to Figure 4.
Author Response
Referee: 1
The study by Galka-Marciniak et al. describes the extent of putative somatic mutations of miRNAs genes in lung cancer (lung adenocarcinoma and lung squamous cell carcinoma). The authors identify that mutations target different functional subregions of miRNAs, and that mature miRNA sequences are more mutated than other regions. These mutations may have a functional impact on the predicted interactions with miRNA targets.
They further experimentally confirm the mutations in hotspot miRNA genes by using an independent cohort. Altogether, this study is sound, timely and could lead to the identification of new diagnostic markers and/or therapeutic targets in lung cancers. Of note, the manuscript is well structured and written. In general, the data are cautiously interpreted. Although the main results are convincing, the study could be improved by addressing these specific comments:
1. In the first part (page 4 and 5), the authors report a tremendous proportion of miRNAs potentially mutated (500 miRNA genes on 1642 miRNA gene regions analyzed). The material and methods section does not clearly indicate if blood samples of the same patients have been analyzed to distinguish single nucleotide polymorphisms from somatic mutation events. Although interesting, the comparison of mutation frequencies in protein-coding genes may not be sufficient to indicate that any detected substitution is a somatic mutation of a noncoding gene. The authors need to address this issue or minor their finding that 500 miRNAs contain somatic mutations.
Ad.1. Thank you for the comment. The mutations identified in our study were annotated as somatic by the standard TCGA pipeline with the use of four different algorithms (MuSE, MuTect2, VarScan2, and SomaticSniper). The algorithms identify somatic mutations (filter out germline variants) comparing variants identified in a cancer sample and in the matched normal (non-cancer) sample. In TCGA each cancer sample was matched with at least one non-cancer sample, either a peripheral blood sample or adjacent histologically-confirmed normal tissue sample or both. In our study, we considered only mutations that passed all standard filters, excluding potentially confusing mutations, e.g, identified by poor quality reads, located in complex/repetitive sequences, or overlapping with known germline SNPs. Additionally, to increase the confidence of detected mutations as somatic variants, we excluded mutations that do not fulfill the following criterion: (i) at least two mutation-supporting reads in a tumor sample (if no mutation-supporting read was detected in the corresponding normal sample); (ii) at least 5x higher frequency of mutation-supporting reads in the tumor sample than in the corresponding normal sample; (iii) somatic score parameter (SSC) > 30 (for VarScan2 and SomaticSniper); and (iv) base quality (BQ) parameter for mutation-supporting reads in the tumor sample > 20 (for MuSE and MuTect2).
To clarify the issue, we added/modified the following pieces of text:
“To identify somatic mutations in miRNA genes, we took advantage of somatic mutation calls performed on WES datasets of 569 LUAD and 497 LUSC paired tumor/normal samples generated within The Cancer Genome Atlas (TCGA) project.”
“We used molecular and clinical data (Level 2) for LUAD and LUSC generated and deposited in the TCGA repository (http://cancergenome.nih.gov). These data included results of somatic mutation calls in WES datasets of 569 LUAD and 497 LUSC samples analyzed against matched normal (non-cancer) samples with the use of the standard TCGA pipeline. The normal (non-cancer) DNA samples were extracted either from peripheral blood samples, adjacent histologically-confirmed normal tissue samples resected at surgery, or from both. The used paired tumor/normal approach permitted the identification and elimination of germline variants.”
2. Figure 1D: It would be interesting to perform the same analysis with somatic mutations of protein-coding genes to see if the distribution of mutations in chromosome 1, 2, 14, 19 is a specific feature of miRNAs or linked to lung cancer.
Ad.2. We agree with the reviewer but at the moment due to the limitation of our permission to TCGA data usage, we are unable to perform analysis of protein-coding genes. However, to address the suggestion, we compared the frequency of mutations in miRNA genes to the frequency of mutations in the entire or protein-coding sequences of different chromosomes reported before for lung as well as for other cancer types e.g. Jager et al. Cell (2013); Hodgkinson et al. Hum Mutat. (2012); Zhou et al. BioRxiv (2017). As it was shown before the overall distribution of somatic mutations between different chromosomes were more even than observed for miRNA genes and therefore does not seem to explain the observed overmutations of miRNA genes in some chromosomes.
To explain the issue, we added an appropriate comment in the discussion section:
“Our results show that mutations are generally spread throughout the genome, even though some chromosomes were overmutated (Chr14 and Chr19) or undermutated (Chr1 and Chr2). Such differences in the mutation load cannot be explained by overall differences of mutation frequency in different chromosomes and were not observed for protein-coding regions [112-114]. As some excess of miRNA-gene mutations at ChrX occurs both in man and woman, it is unlikely to be caused by the observed before hypermutation of the inactive ChrX [112].”
3. Figure 3: The functional impact of miRNA mutation through target gene regulation may result in changes in signalling pathways. This can be addressed by performing pathway enrichment analyses of the predicted target genes in both conditions (wt and mutated miRNAs). If specific pathways are enriched, their relevance in lung cancer biology may be discussed. This will reinforce the idea that somatic mutation in miRNAs may have a functional biological impact in lung cancer.
Ad.3. We agree with the comment. We considered this type of analysis before (e.g. with the use of the Panther classification system or TAM 2.0 tool for miRNA set analysis). However, we desisted the analysis as the results were mostly inconclusive. It was probably mostly due to (i) low fraction of somatic mutations that are actually functional (most somatic mutations occur by chance), and (ii) low specificity of the target prediction algorithms (only a small fraction of predicted targets are functional). We believe that such an analysis would be more informative if performed on validated targets of confirmed driver mutations.
Minor comments:
4. Figure 2 and 4: another colour code could be used to better highlight the differences in duplex positions and seed regions.
Ad.4. We agree. We have modified Figure 2, Figure 4, and Supplementary Figure S2 to make seed regions more visible.
5. Data presented page 15 on the experimental validation of hotspot miRNA mutations by Sanger sequencing could be moved to Figure 4.
Ad.5. The mutations detected in the experimental analysis are shown in Figure 4 (orange symbols). To clarify the issue, we have added the reference to Figure 4 in the paragraph describing experimental validation. Please, note that only mutations localized in the pre-miRNA precursor sequence and the flanking 25-nt 5’ and 3’ sequences are shown in this figure.
Reviewer 2 Report
Galka-Marciniak and colleagues performed a valuable study focusing on mutations in miRNA genes in adenocarcinomas and squamous-cell carcinomas of the lung using TCGA derived datasets. They provide interesting results concerning the location of mutations within the miRNA structure and predicted effects on the molecular level. There are a few minor issues to.
MINOR:
Please give in the whole manuscript including also the figure legends the exact number of analyzed samples and/or identified mutations. Roughly estimated numbers as for example 500 LUAD and 500 LUSC are not good science.
Indicate in the 4.3 Statistics section the original reference from Benjamini et al. and not the link for a handbook homepage.
The results presented in reference 112, that miR-205 acts an oncogene in squamous-cell lung carcinomas is an important piece of information and strengthen even more the finding of two times identical mutations in the miR-205 gene in the LUSC cohort. This have to be discussed and highlighted in the results and the discussion section.
Since no scripts are included in the supplementary files for download, please include in the methods paragraph at least the sentence: Scripts are available upon request.
Author Response
Referee: 2
Galka-Marciniak and colleagues performed a valuable study focusing on mutations in miRNA genes in adenocarcinomas and squamous-cell carcinomas of the lung using TCGA derived datasets. They provide interesting results concerning the location of mutations within the miRNA structure and predicted effects on the molecular level. There are a few minor issues to.
MINOR:
1. Please give in the whole manuscript including also the figure legends the exact number of analyzed samples and/or identified mutations. Roughly estimated numbers as for example 500 LUAD and 500 LUSC are not good science.
Ad.1. We have indicated the exact number of the analyzed samples and identified mutations in the revised version of the text and the Figure 5 legend.
2. Indicate in the 4.3 Statistics section the original reference from Benjamini et al. and not the link for a handbook homepage.
Ad.2. As suggested, the reference Benjamini et al. (1995) was added to the manuscript.
3. The results presented in reference 112, that miR-205 acts an oncogene in squamous-cell lung carcinomas is an important piece of information and strengthen even more the finding of two times identical mutations in the miR-205 gene in the LUSC cohort. This have to be discussed and highlighted in the results and the discussion section.
Ad.3. We agree with this comment and to address this issue we included the following sentences to the results and discussion sections:
“More than half of the mutated miRNA genes are annotated as high confidence in miRBase. Among them are genes of miRNAs well known to play an important role in cancer development. An example is the miR-205 gene, one of the most intensively studied cancer-related miRNA acting both as oncomiR and suppresormiR (reviewed in [58]).”
“Examples of the mutated suppressormiR genes may be the Let-7a-2, let-7c, and let-7e genes belonging to the let-7 family, known to have tumor suppressor function by repressing cell proliferation and regulating the cell cycle [112] and the miR-205 gene, which may act either as suppresormiR inhibiting cancer progression by suppressing EMT or oncomiR accelerating cancer development by facilitating tumor initiation and proliferation [58,113]. It was shown that miR-205 [24,25] belongs to the group of miRNAs most commonly overexpressed and that its gene is frequently gained or amplified in lung cancer [33]. The same n.35C>T mutation in the miR-205 seed sequence was observed in two samples in our studies (Figure 3 and Supplementary Figure S3B). As shown in Figure 3 the mutation causes the loss of many miR-205 targets, including well-validated MED1, ERBB3, VEGFA, E2F1, and PTEN, that are crucial for the cancer-related functions of miR-205. Less effective downregulation of the tumor suppressor PTEN may result in tumor suppression whereas less effective downregulation of such oncogenes as VEGFA (the key mediator of angiogenesis) or E2F1 (the transcription factor controlling cell cycle) may enhance tumor progression, invasion, and/or metastasis.”
4. Since no scripts are included in the supplementary files for download, please include in the methods paragraph at least the sentence: Scripts are available upon request.
Ad.4. We agree with the reviewer’s comment. All our scripts are submitted and are freely available at the GitHub repository. The appropriate information is in the text.
“The extraction was performed with a set of in-house Python scripts (available at https://github.com/martynaut/mirnaome_somatic_mutations).”
Reviewer 3 Report
The paper analyze the somatic mutations on microRNA sequences in NSCLC. The authors use a bioinformatic rigorous approach to dissect the position of the mutations in particular in a subset of hotspot microRNAs.
The manuscript is well done and descriptive, and the authors took in consideration the most important variables of the case.
Author Response
Referee: 3
The paper analyze the somatic mutations on microRNA sequences in NSCLC. The authors use a bioinformatic rigorous approach to dissect the position of the mutations in particular in a subset of hotspot microRNAs.
The manuscript is well done and descriptive, and the authors took in consideration the most important variables of the case.
We would like to thank the reviewer for the comment and the effort and time devoted to reviewing our manuscript.
Reviewer 4 Report
This manuscript describes somatic mutations in miRNA genes in two types of lung cancer by analysis of sequencing data. The sequencing data covered 1642 miRNA genes. 1000 mutations were found in about 500 miRNA genes in about 1000 tumor samples. Thus, the mutation frequency is about one mutation in 1642 miRNA genes. The authors claim that the number of miRNA genes mutated in these two cancers significantly exceeded the number expected by chance. It is unknown how they calculated the expected mutation frequency. Cells in early tumor development stage generally grow very fast with genome instability. The spontaneous mutation frequency in this stage is very high. The mutation frequency may be higher than 1 in 1642 genes. Therefore, it is possible that the mutations in the miRNA genes occurred spontaneously rather than produced as the “driver mutations”. The authors should provide a control group indicating the spontaneous mutation frequency in the lung cancer tissues.
Also, if you found 1000 mutations in 500 different miRNA genes, it is hard to say they are driver mutations. Driver mutations are located at a few of genes related to carcinogenesis. It is impossible that there are so many miRNA genes involving in carcinogenesis. The authors need performing wet experiments to show mutations in the miRNA genes can “drive” cells into tumors.
Author Response
Referee: 4
1. This manuscript describes somatic mutations in miRNA genes in two types of lung cancer by analysis of sequencing data. The sequencing data covered 1642 miRNA genes. 1000 mutations were found in about 500 miRNA genes in about 1000 tumor samples. Thus, the mutation frequency is about one mutation in 1642 miRNA genes. The authors claim that the number of miRNA genes mutated in these two cancers significantly exceeded the number expected by chance. It is unknown how they calculated the expected mutation frequency.
Ad.1. It is a misunderstanding. In the paper, we do not claim “that the number of miRNA genes mutated in these two cancers significantly exceeded the number expected by chance”. We showed that the overlap of miRNA genes mutated in both cancers is higher than expected by chance. The analysis was performed with the use of a hypergeometric probability test (as used before; e.g., Abou-Khalil B et al. Nat Commun, 2018; Luo W et al. Science Adv, 2018; Goetsch PD et al. Plos Genet, 2017; Rohlfing A Plos Genet, 2011). The test showed a 2.3x enrichment of miRNA genes mutated in both cancers (p<1 × 10-45). To clarify this issue, we modified the misleading sentence. The revised sentence is:
“As shown in Figure 1B, the sets of mutated miRNA genes strongly overlapped between the two types of cancer, and the number of miRNA genes mutated in both cancers significantly exceeded the number expected by chance (if the occurrence of mutations in both cancers would be independent; hypergeometric probability with normal approximation test: fold enrichment=2.3; p<1 × 10-45).”
2. Cells in early tumor development stage generally grow very fast with genome instability. The spontaneous mutation frequency in this stage is very high. The mutation frequency may be higher than 1 in 1642 genes. Therefore, it is possible that the mutations in the miRNA genes occurred spontaneously rather than produced as the “driver mutations”. The authors should provide a control group indicating the spontaneous mutation frequency in the lung cancer tissues. Also, if you found 1000 mutations in 500 different miRNA genes, it is hard to say they are driver mutations. Driver mutations are located at a few of genes related to carcinogenesis. It is impossible that there are so many miRNA genes involving in carcinogenesis. The authors need performing wet experiments to show mutations in the miRNA genes can “drive” cells into tumors.
Ad.2. We absolutely agree with the reviewer. It was not our intention to suggest that all or most of the detected mutations are functional or driver mutations. We agree with the reviewer that only very small fraction of somatic mutations may be considered as functional. As potentially functional, for example, may be considered some of the mutations identified in hotspot miRNA genes. Still as advised in the paper the result should be interpreted very cautiously. The prove that particular variant or variants function as cancer drivers require much more work, including functional assays and/or additional computational/statistical analysis of the different group of cancer samples, that are beyond the scope of our study. Nonetheless, we believe that statistical/descriptive characterization of somatic mutations in particular cancers provides valuable information and has sense. It is the important first step in understanding the role of somatic variation in non-coding miRNA genes (as suggested, e.g., by reviewer 1). For example most of TCGA preliminary results published in such journals as Cell or Nature are based on statistical/descriptive analysis of somatic mutations occurring in exoms of particular cancers (we show similar analysis in miRNome), even though, as rightly stated by the reviewer most of the detected mutations are spontaneous – passenger mutations randomly occurring throughout the cancer genome. These whole-exome or whole-genome analyses also show that the number of spontaneous mutation (background) differs substantially (orders of magnitude) even among samples from the same cancer type (Bailey et al. Cell 2018).
To clarify this issue, we added an appropriate comment to the discussion:
“In this study, we detected 1091 mutations in 521 miRNA genes (Figure 1A) and showed that approximately 50% of the analyzed cancers had at least one mutation in at least one miRNA gene. It has to be noted, however, that similarly, as in protein-coding genes, the great majority of mutations detected in miRNA genes constitute spontaneous mutations accumulating during cancer development and only a small fraction of these variants may be a potential driver or functional mutations.”
We also indicate this in other parts of the text:
Results:
“However, due to the low number of mutations in particular hotspot miRNA genes, the abovementioned associations (not corrected for multiple comparisons) must be interpreted cautiously and cannot be generalized without further validation.”
Discussion:
“Most of these alterations are neutral (passenger) changes occurring randomly throughout the genome, but others may be cancer-driving mutations. The accumulation of such mutations in a particular gene or the recurrent occurrence of a particular mutation may indicate its importance in cancer development.”
“Nevertheless, the statistical power of these analyses was low (low number of mutations in a particular gene), and therefore, the determination of whether the identified hotspot miRNA genes have cancer-driver potential requires further statistical and experimental validation. On the other hand, even if the detected mutations are not primary drivers, they may still play a role in cancer. It was recently shown that passenger mutations may both prevent further cancer progression [123] and cooperate in driving cancer development and drug resistance [124,125].”
“Therefore, our results, as well as the recently published tool ADmiRE [34], may help to prioritize potentially functional mutations and to develop a better algorithm for the identification of driver mutations in regions coding for non-coding RNAs, particularly miRNAs.”
Round 2
Reviewer 1 Report
My concerns have been adequately addressed and I don't have further comments.
Author Response
Thany you for the review.
Reviewer 4 Report
The authors did make an effort for analysis of these sequencing data in miRNAs
Author Response
We would like to thank the reviewer for the effort and time devoted to this revision and for raising the scores of our manuscript. We believe that changes suggested by the Academic Editor concerning the replacement ofthe word ‘mutation’ with more neutral phrases that do not implicate the functional consequence of the identified somatic variants are also in line with the reviewer comments.